# BLENDING DIVERSE PHYSICAL PRIORS WITH NEURAL NETWORKS

## ABSTRACT

Rethinking physics in the era of deep learning is an increasingly important topic. This topic is special because, in addition to data, one can leverage a vast library of physical prior models (e.g. kinematics, fluid flow, etc) to perform more robust inference. The nascent sub-field of *physics-based learning* (PBL) studies this problem of blending neural networks with physical priors. While previous PBL algorithms have been applied successfully to specific tasks, it is hard to generalize existing PBL methods to a wide range of physics-based problems. Such generalization would require an architecture that can adapt to variations in the correctness of the physics, or in the quality of training data. No such architecture exists. In this paper, we aim to generalize PBL, by making a first attempt to bring neural architecture search (NAS) to the realm of PBL. We introduce a new method known as physics-based neural architecture search (PhysicsNAS) that is a top-performer across a diverse range of quality in the physical model and the dataset.

## 1 INTRODUCTION

Advances in machine learning can transform the way physical calculations are performed. Many physical models are idealized and do not precisely match real-world data. An elementary example would be equations for projectile motion which do not account for air resistance. Using these idealized equations, a completely *physics-driven approach* would have large errors on real-world data. A separate approach is completely *data-driven*, e.g., one could repeatedly record real-world projectile tosses and use a regression model to estimate a future trajectory; unfortunately, this approach requires large datasets and lacks interpretability. To bridge this gap, the field of *physics-based learning* (PBL) aims to blend physical priors with data-driven inference, to combine the best of both worlds.

Previous PBL architectures have achieved competitive performance on a wide variety of tasks in microscopy (Rivenson et al., 2019; Nehme et al., 2018; Nguyen et al., 2018; Sinha et al., 2017; Goy et al., 2018), low level and high level computer vision (Ba et al., 2019; Sun et al., 2019), medical imaging (Jin et al., 2017; Kang et al., 2017), and robot control (Zeng et al., 2019; Ajay et al., 2019). These seemingly diverse problem statements share a common thread: the presence of a partially known physical prior that can be blended with a neural network.

Unfortunately, existing PBL methods are typically designed for a specific task. Generalization would (as a first step) require a PBL architecture capable of adapting to variations in the correctness of physics or the quality of training data. Our experiments show that no such architecture exists (Figure 1 and Section 4.3). Having a general recipe for blending physics and learning is an important step in adopting physics-based learning to encompass the wide range of physical problems, where priors are only approximate and training data can be sparse.

In this work, we approach the problem of PBL from a different angle. Inspired by work in neural architecture search (NAS) (Zoph & Le, 2016; Baker et al., 2016; Liu et al., 2018; Cai et al., 2018), we propose a first attempt to automatically find the optimal PBL architecture, taking into account characteristics of not just data, but also physics. To incorporate physical models into NAS, we find that three modifications must be made to the existing NAS framework: (1) the inclusion of physical inputs; (2) the inclusion of physical operation sets; and (3) edge weights to normalize variations in the degrees of freedom introduced by the inclusion of physical operators. As these modifications are specific to the PBL problem, we refer to our algorithm as PhysicsNAS. As shown in Figure 1, the

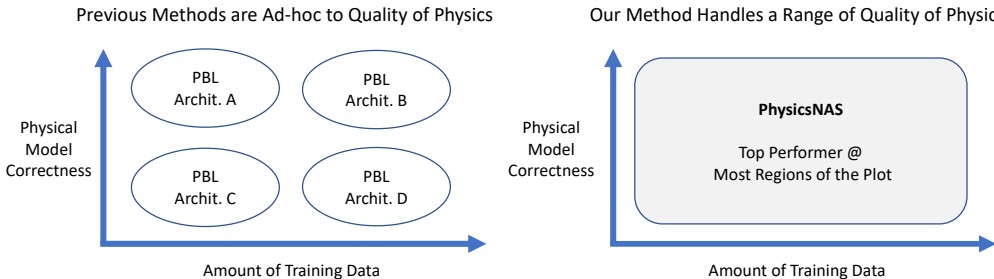

Figure 1: **Generalizing PBL across a range of sparsity in training data and correctness in the physical model.**

goal of PhysicsNAS is to handle a diverse range of quality in the physical prior or data. Experiments in Section 4.3 offer support for this goal, where PhysicsNAS outperforms previous PBL methods on multiple physical tasks across a range of physical prior and dataset conditions. The performance improvement over existing PBL methods ranges from 3% to 60%.

Our contributions to physics-based learning can be summarized as follows:

- We make a first attempt at bringing neural architecture search (NAS) into the realm of physics-based learning (PBL), introducing PhysicsNAS as a new method for PBL.
- We show in our experiments that PhysicsNAS generalizes to a wider range of diversity in data and physical priors, as compared to previous PBL methods;
- We interpret the converged architectures of PhysicsNAS in context of prior PBL work, to provide general evidence that: (a) Accurate physical operations can be embedded into the network design if there is enough training data available; (b) Residual Physics is a preferred alternative to inaccurate physical operations; (c) Physical Fusion is a general strategy that can be adopted in various physical environments. Separate from our work, these insights can help lay a foundation for how to explain the choice of PBL models in future work.

Although our primary contributions are to PBL, it is worth noting that conventional differentiable NAS approaches (Liu et al., 2018) are not designed to incorporate physical priors; in developing this paper we found it necessary to modify such approaches to incorporate physical priors as both inputs and candidate operations.

## 2    CATEGORIZING PRIOR WORK IN PHYSICS-BASED LEARNING

There has been remarkable progress in blending physical priors with neural networks, over the past few years. Here, we make a first attempt to group previous methods into the four categories as illustrated in Figure 2:

- **Physical Fusion** feeds the solution from physics-based models as part of the input (Karpatne et al., 2017; Ba et al., 2019). The solutions can be stacked with the original input, or additionalidentical network branches can be used to extract features separately;
- **Residual Physics** is another way to improve the model-based solutions with deployments in robot control (Zeng et al., 2019; Ajay et al., 2019) and medical imaging (Jin et al., 2017; Kang et al., 2017). By adding the physical solution onto the network output, the neural networks only need to learn the mismatch between the model-based solution and the ground truth in this case;
- **Physical Regularization** harnesses the regularization term from a set of physical constraints to penalize the network solutions. The regularization term can be appended as part of the loss function explicitly (Karpatne et al., 2017; Stewart & Ermon, 2017; Raissi et al., 2017; Raissi, 2018; Fei et al., 2019), or through a reconstruction process from physics (Che et al., 2018; Chen et al., 2018; Pan et al., 2018);
- **Embedded Physics** takes the physical model inside the network optimization loop, where the physical model acts as a skeleton, and the network is in charge of learning parameters used in these

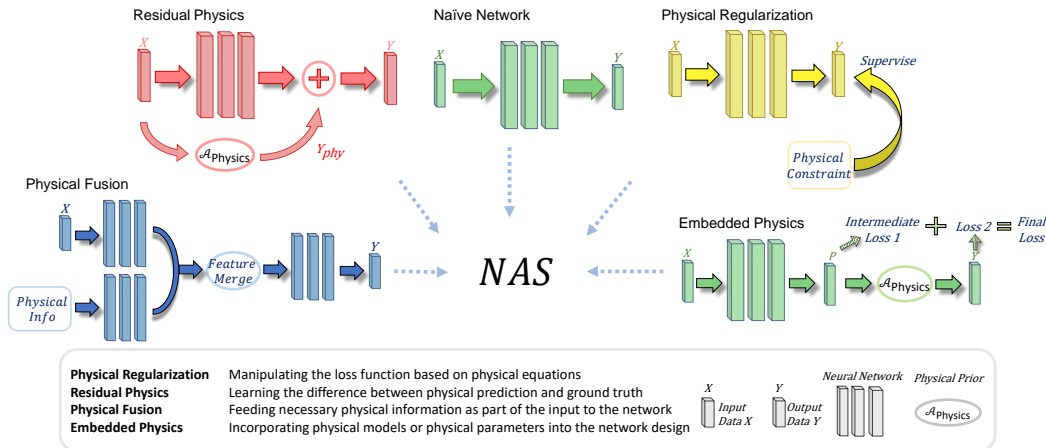

Figure 2: **An overview of proposed NAS-based blending approach.** Our PhysicsNAS takes advantage of all the existing methods on blending physical prior, and is capable of generating new hybrid architectures for tasks under diversified physical environments. With the augmented search space and knowledge from prior information, it is possible for the proposed PhysicsNAS to generalize its performance with limited number of training samples.

models. Unrolled networks (Gregor & LeCun, 2010; Diamond et al., 2017; Kellman et al., 2019; Monakhova et al., 2019; Sitzmann et al., 2018), PDE-Nets (Long et al., 2017), and variational networks (Hammernik et al., 2018; Chakrabarti, 2016) can all be classified into this category. During training, auxiliary intermediate losses can be inserted to guarantee the learned parameters indeed carry their corresponding physical meanings as well (Hui et al., 2019; Song & Funkhouser, 2019; Li et al., 2019).

Continuing to propose new models for PBL is a viable direction, however this may not address adaptability to diverse scenarios of physical model mismatch and sparsity in training data. PhysicsNAS is a different tack, where we design basic operation sets inspired by PBL strategies, and allow networks to customize their architectures during training.

## 3 PHYSICSNAS

In what follows, we describe the PhysicsNAS algorithm. In Section 3.1, we discuss the problem setup. We then describe the search algorithm in Section 3.2 and the detailed features of PhysicsNAS in Section 3.3.

### 3.1 PROBLEM SETUP

In the PBL problem, we have access to a training set $D_{train} = \{(x_i, y_i)\}_{i=1}^N$ and a partially known physical operator $\mathcal{A}_{phy}$. Each sample within the training set is a data pair $(x_i, y_i)$ formed by an input instance $x_i \in X$ and the corresponding output (label) $y_i \in Y$, and the objective is to learn a function $f(\cdot)$ that maps input space to output space $X \to Y$. $f(\cdot)$ is approximated by a physics network from a search space $\mathcal{H}$ with hypotheses $\hat{f}(\omega, \alpha, \mathcal{A}_{phy})$, where $\omega$ denotes network parameters and $\alpha$ denotes architecture parameters. The learning algorithm searches inside $\mathcal{H}$ and tries to find $\{\omega, \alpha\}$ that parameterizes the optimal $\hat{f}(\omega, \alpha, \mathcal{A}_{phy})$ for $D_{train}$. The challenge for these problems lies in finding a suitable method to incorporate $\mathcal{A}_{phy}$ into the network design under diverse regimes of physical model mismatch.

### 3.2 SEARCH ALGORITHM

We develop PhysicsNAS based on differentiable NAS techniques (Liu et al., 2018; Cai et al., 2018). With learnable architecture parameters $\alpha$ and continuity relaxation, both network architectures and

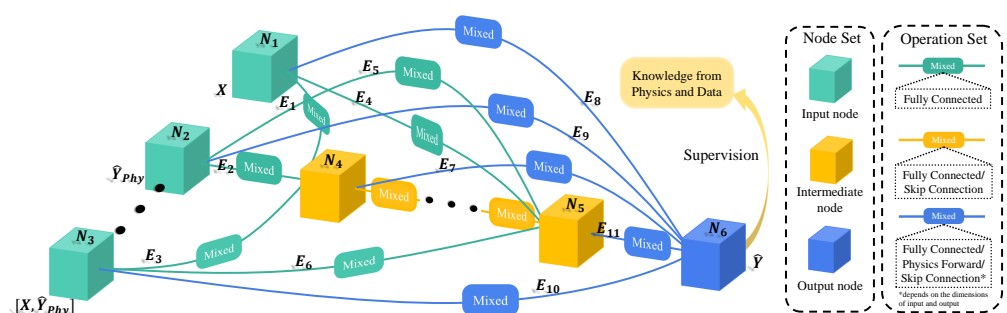

Figure 3: **Search space of our PhysicsNAS.** In the proposed PhysicsNAS, all the nodes are densely connected by mixed operators from predefined candidate operation sets. The hidden nodes can obtain information from the original inputs or from previous hidden nodes within this search setup. The training process is supervised by both ground truth and physical constraints.

parameters can be updated using gradient descent. In contrast to NAS for complicated vision tasks, we do not search the cell structures and apply these searched cells to predefined meta-architectures in PhysicsNAS. As such, PhysicsNAS tries to learn an architecture that links the network input and network output directly. The search space of PhysicsNAS is illustrated in Figure 3, where the whole architecture is represented by a directed acyclic graph with nodes $\{N_i\}_{i=1}^N$ and edges $\{E_m\}_{m=1}^M$. Each edge connects two nodes $(N_i, N_j)$ through a mixed operator, and each node corresponds to a type of input or a feature vector extracted from previous nodes through the mixed operators. The output of the mixed operator between $(N_i, N_j)$ is the gated sum of all candidate operations $\{o_k\}_{k=1}^K$:

$$m_{ij}(n_i) = \sum_{k=1}^{K} g_{o_k} o_k(n_i),\qquad(1)$$

where $m_{ij}$ is the output of this mixed operator, $n_i$ is the feature vector of node $N_i$, $g_o$ is the binarized operation mask based on the softmax probability of architecture parameters $\alpha_o$ in (Cai et al., 2018), and $K$ is the number of operations inside a edge, which depends on the properties of node pair $(N_i, N_j)$. The nodes are densely connected, so that $n_j$, the output features of node $N_j$, is the gated sum of features from all its previous nodes:

$$n_j = \sum_{i=0}^{j-1} g_{e_i} m_{ij}(n_i) = \sum_{i=0}^{j-1} g_{e_i} \sum_{k=1}^{K} g_{o_k} o_k(n_i),\qquad(2)$$

where $g_e$ is the binarized edge mask based on the softmax probability of architecture parameters $\alpha_e$, and $N_j$ can either be an intermediate node or the output node.

During training, we retain two incoming edges for each node and one operation for each edge through the binary gate sampling in (Cai et al., 2018). While for inference, we pick two candidate edges with the largest edge probabilities, and select the operation with maximum operation probability for each of the two edges. We choose two edges for each node to leave the potential for PhysicsNAS to learn complicated structures, like skip connection and multi-stream encoding. In order to learn both the network weights and the associated architecture parameters, we update these two sets of parameters alternately. In the architecture step, we freeze the network weights $\omega$ and minimize the validation loss $\mathcal{L}_{val}(\omega, \alpha)$ by updating $\alpha$. In the network step, we update $\omega$ to minimize the training loss $\mathcal{L}_{train}(\omega, \alpha)$ with frozen $\alpha$.

### 3.3 PhysicsNAS Features

To incorporate priors into existing differentiable NAS (Liu et al., 2018; Cai et al., 2018), we make three unique modifications into the search process of PhysicsNAS.

**Physical Inputs** As a first step in blending physics into PhysicsNAS, we need to prepare unique input nodes that take into account four categories of input information: 1) the data input $X$; 2) the duplicated data input $X_{dup}$ to verify whether physical information is indeed necessary since each

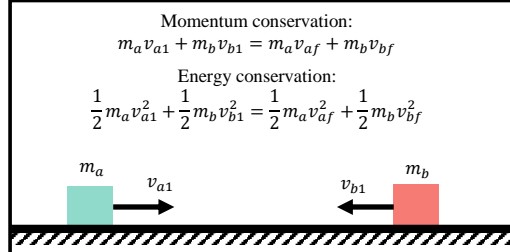

Figure 4: **We evaluate our method on a simulator of classical tasks.** The first task (Left) is predicting the trajectory of a ball being tossed, and the second task (Right) is estimating the velocities of two objects after collision.

node has to pick two edges; 3) the estimated solution from physics $\hat{Y}_{phy} = \mathcal{A}_{phy}(X)$; and 4) the concatenation of $X$ and $\hat{Y}_{phy}$ to test which stage to conduct the physical fusion.

**Physical Operations** To merge physical models inside the network, we create physics-informed operation sets $O = \{o_{NN_1}, ..., o_{NN_L}, o_{phy}\}$, where $o_{NN_i}$ denotes the neural network operations (e.g.,fully-connected layer, skip connection) and $o_{phy}$ denotes the physical forward operation. Specifically, for the physical forward module, we also use a light-weight network layer, such as a fully-connected (FC) layer, to make the size of its input consistent with the parameter size required by the physical module. Physical forward modules are only included in the edges that connect to the output node in our implementation.

**Edge Weights** In PhysicsNAS, not all edges are created with the same amount of operations, since they are used to connect different types of nodes. Consequently, if we select the edges purely based on the operation probabilities, edges with fewer operations are naturally preferred due to the softmax probability, which causes a biased architecture selection. We solve this issue by introducing the edge weight as described in Equation 2. After searching, we first pick a desired edge according to the edge weights, and then select the desired operation for that edge based on the operation weights.

## 4 EXPERIMENTS AND RESULTS

To comprehensively evaluate PhysicsNAS, we simulate two representative physical tasks for which we can vary the model mismatch: 1) predicting trajectories of an object being tossed; and 2) estimating the speed of rigid objects after collision. Figure 4 illustrates these tasks; further details are provided in Section 4.1. Comparison PBL architectures are described in Section 4.2. Finally, we evaluate PhysicsNAS and provide a detailed analysis on the searched architectures in Section 4.3.

### 4.1 DESCRIPTION OF TASKS

For the TOSSING TRAJECTORY PREDICTION task (see Figure 4 for visualization), the initial three locations of the object $\{l_1, l_2, l_3\}$ are given as input $X$, and our objective is to predict locations of this object in the following 15 time stamps, $\{l_4, l_5, ..., l_{18}\}$. We only consider the displacement within a 2D plane, therefore, the coordinates of each location can be represented by two numbers, i.e. $l_i = (l_{x_i}, l_{y_i})$. We adopt the following elementary free-falling equations as the prior, and examine different methods under this inadequate physical prior:

$$\hat{Y}_{phy} : \begin{cases} l_{x_i} = l_{x_1} + v_x t_i \\ l_{y_i} = l_{y_1} + v_y t_i - \frac{1}{2} g t_i^2 \end{cases}, \tag{3}$$

where $l_{x_i}$ and $l_{y_i}$ denote the object location at time $t_i$, $l_{x_1}$ and $l_{y_1}$ are the initial location of the object, $v_x$ and $v_y$ denote the initial velocities along horizontal and vertical directions respectively, and $g$ is the fixed gravitational acceleration of $9.8 m/s^2$. We introduce two model mismatches: the random acceleration as the winds and an additional damping factor based on $F_{air} = k \times v^2$ to simulate the air resistance. The future locations estimated according to mismatched prior are used as the physical input $\hat{Y}_{phy}$. As to the physical modules in Embedded Physics and PhysicsNAS, we

estimate parameters $\{\hat{l}_{x_1}, \hat{l}_{y_1}, \hat{v}_x, \hat{v}_y\}$, and substitute these parameters into Equation 3 as the physical operation.

In the COLLISION SPEED ESTIMATION task (see Figure 4 for visualization), we use the speed of two objects at the initial two time stamps, object mass, and the distance between these two objects $\{v_{a_1}, v_{a_2}, v_{b_1}, v_{b_2}, m_a, m_b, D\}$ as the input $X$ to estimate the speed after their collision $\{v_{a_f}, v_{b_f}\}$. We assume the objects have the different mass, and can only move along one direction. Based on energy conservation and momentum conservation for perfectly elastic collision, we adopt the following prior:

$$\hat{Y}_{phy} : \begin{cases} v_{a_f} = \frac{1}{m_a+m_b}[v_{a_1}(m_a - m_b) + 2m_b v_{b_1}] \\ v_{b_f} = \frac{1}{m_a+m_b}[v_{b_1}(m_b - m_a) + 2m_a v_{a_1}] \end{cases} . \tag{4}$$

We add sliding friction to the system as intentional model mismatch, where conservation prior in Equation 4 does not hold. Solutions without the consideration of friction are used as $\hat{Y}_{phy}$, and $\{\hat{m}_a, \hat{m}_b, \hat{v}_{a_1}, \hat{v}_{b_1}\}$ are estimated for the physical modules.

## 4.2 MANUALLY DESIGNED PBL METHODS

For the sake of comparison, several manually designed architectures from Section 2 are also evaluated. We use a three-layer multilayer perceptron (MLP) as the naive data-driven baseline, since it has sufficient expressiveness to fit any continuous function, especially the elementary physical tasks we have chosen (Csáji, 2001). Network structures for the Physical Regularization model and the Residual Physics model are the same as the naive model. The output of the Residual Physics model is a summation of $\hat{Y}_{phy}$ and the learned residual from the network, while there is an additional regularization term in loss function of the Physical Regularization model. Since only a partially correct physical prior is used, directly using physical solution as the regularization will in turn aggravate the error. Thus, we introduce an ReLU-based regularization similar to (Karpatne et al., 2017). The regularization loss penalizes the network solution based on the assumption that the object moves along one direction in the horizontal axis for the trajectory prediction task, and the total kinetic energy is less than the initial kinetic energy for the speed estimation task. In the Physical Fusion approach, two separate branches are utilized to extract features from $X$ and $\hat{Y}_{phy}$ respectively, and each of them is a two-layer MLP. The extracted features will then be concatenated and fed into the output layer. The Embedded Physics model first estimates necessary parameters in Equation 3 and Equation 4 with a three-layer MLP, and then produces trajectory estimation based on the fixed physical process. All the above models are supervised by the ground-truth future locations with mean square error (MSE) loss, and the hidden dimension for FC layers are 128.

## 4.3 RESULTS ANALYSIS

**Training Details** To evaluate the importance and success of the proposed approach, we vary the physical model mismatch and sparsity in training data in a controlled manner. When training PhysicsNAS, we split the training set into two subsets of the same size to update architecture variables and network variables respectively. We limit the number of learnable nodes in PhysicsNAS to be 5, and retrain the searched architectures with full training sets after searching. The models are implemented in PyTorch (Paszke et al., 2017), and are trained using the Adam optimizer (Kingma & Ba, 2014). Moreover, for all baseline approaches we compare in this paper, we fine tune their hyperparameters in order to make fair comparisons. We choose three hyperparameter sets for each scenario and run five times for each method. We finally pick out the best result for each method.

**Performance Comparison** We apply the proposed PhysicsNAS to learn architectures embedded in the search space. The testing results of PhysicsNAS and other existing PBL methods (as detailed in Section 4.2) are summarized in Table 1 and Table 2. As shown in these two tables, the performance of different PBL models varies based on the disparity of mismatch levels and training data sizes, while PhysicsNAS is capable of generating architectures that outperform these manual PBL models consistently. Results in Figure 5 further demonstrate this capability along data dimension and physics dimension in a fine-grained scale. We also conduct an ablation study on PhysicsNAS by removing task-specific adaptations such as physical inputs and operations. Results in Appendix D show the task-specific adaptations improve the performance of PhysicsNAS at different mismatch levels, demonstrating that having physical inputs and physical operations in PhysicsNAS's search

| Mismatch Level | Low | | High | |
|---|---|---|---|---|
| Sample Amount | 32 | 128 | 32 | 128 |
| Naive Network | 0.684 | 0.232 | 0.696 | 0.267 |
| Physical Fusion | **0.266** | 0.200 | **0.321** | **0.178** |
| Residual Physics | 0.323 | 0.192 | 0.481 | 0.279 |
| Embedded Physics | 0.617 | **0.169** | 0.617 | 0.305 |
| Physics Reg. | 0.459 | 0.272 | 0.674 | 0.315 |
| PhysicsNAS | **0.183** | **0.097** | **0.264** | **0.152** |

Table 1: **Testing performance on tossing task.** We adopt the average Euclidean distance between the ground truth and the predicted locations as the evaluation metric (lower distance is better). The low mismatch level corresponds to a small random initial acceleration range $[-1m/s^2, 1m/s^2]$ and a small damping factor 0.2. The high mismatch level corresponds to a large acceleration range $[-3m/s^2, 3m/s^2]$ and a large damping factor 0.5. The best model is marked in **red** and the sub-optimal is in **blue**.

| Mismatch Level | Low | | High | |
|---|---|---|---|---|
| Sample Amount | 32 | 128 | 32 | 128 |
| Naive Network | 1.974 | 0.319 | 8.865 | 4.534 |
| Physical Fusion | **0.868** | 0.174 | **6.892** | 4.596 |
| Residual Physics | 1.053 | **0.173** | 11.750 | 5.859 |
| Embedded Physics | 2.105 | 0.258 | 7.962 | 4.546 |
| Physics Reg. | 1.916 | 0.271 | 8.631 | **4.451** |
| PhysicsNAS | **0.724** | **0.121** | **6.741** | **4.157** |

Table 2: **Testing performance on collision task.** We use similar Euclidean distance between the estimated speed and the ground-truth speed as the metric (lower distance is better). The low mismatch level corresponds to a random initial friction coefficient in range [0.28, 0.32], and the high mismatch level corresponds to a friction coefficient in range [0.45, 0.55]. The best model is marked in **red** and the sub-optimal is in **blue**.

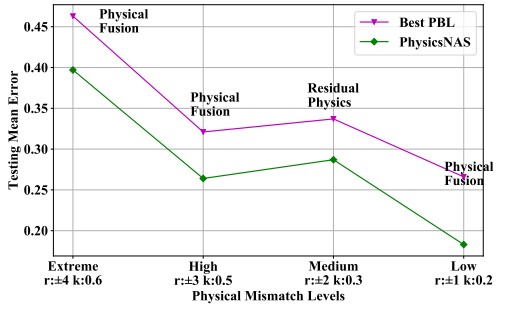
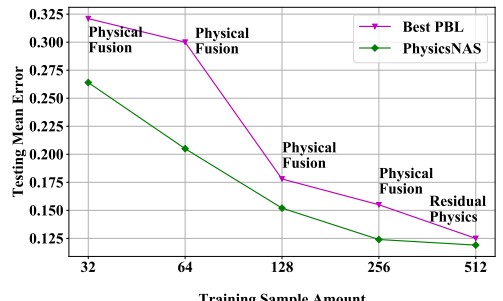

Figure 5: **PhysicsNAS has lower errors compared with the best PBL methods over a range of quality conditions in physics and data.** The left figure shows comparison between best PBL methods and PhyisicsNAS along different physical mismatch levels. The physical mismatch levels are from extreme to low, respectively. Here $(r : \pm i, k : j)$ refers to the mismatch level of a initial acceleration range $[-im/s^2, im/s^2]$ and a damping factor $j$. Analogously, the right figure shows comparison along different data amounts. Results in the left figure are all trained with 32 samples and the results in the right figure are trained at low physical mismatch level. The plots show error; lower curves are preferred.

space is necessary. Our experiments also show that PhysicsNAS is able to perform inference on small training datasets: the physical prior reduces the demand for high-fidelity training samples. We find that PhysicsNAS only requires less than 64 training samples to reach the same testing performance of a naive MLP with 256 training samples. Moreover, the performance gap between PhysicsNAS and naive MLP method minimizes as the number of training samples increases. This suggests that PhysicsNAS is more favorable in scenarios where training data is not rich enough on the other hand. A detailed discussion about how PhysicsNAS reduces the demand on training data is provided in Appendix E.

**Utilization of Physical Inputs** The physical inputs are always selected in our searched architectures for these two tasks, which verifies the importance of physical information during learning. Please refer to Appendix B for the illustration of a range of searched architectures corresponding to scenarios in Table 1 and Table 2.

**Utilization of Physical Operations** The selection of physics-inspired operations are more nuanced, depending on the accuracy of physical information encoded in the physical operations as well as the amount of training data. Figure 6 shows two examples, one where physical operations are selected and the other where they are not selected. In particular, the inaccurate physical operation in the

trajectory task is preferred at early training epochs. However, as training proceeds, the learned FC modules achieve higher accuracy and the network thus discards the physical operations. As a result, the Residual Physics strategy is adopted in the final searched architecture. For the collision task, the physical operation could model the perfectly elastic collision completely, and the estimated physical solutions are precise if the estimated physical parameters are accurate. Therefore, physical operations are selected when there exists a robust estimation of physical parameters. However, it usually requires sufficient training samples to obtain this robust estimation, which might be a reason why physical operations are only selected in the cases with 128 training samples. More details about the changes of network architectures during searching can be found in Appendix F.

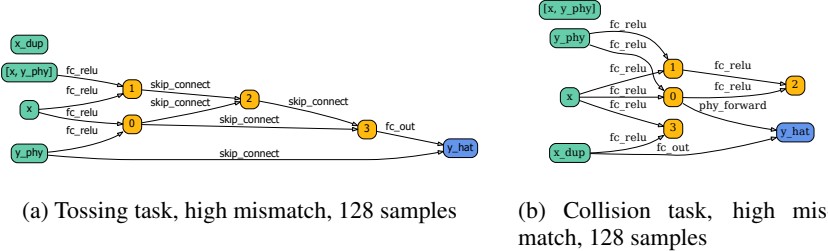

(a) Tossing task, high mismatch, 128 samples     (b) Collision task, high mismatch, 128 samples

Figure 6: **Utilization of physical operations in PhysicsNAS.** The selection of physics-inspired operation depends on its accuracy. PhysicsNAS tends to utilize the physical operations when they are more accurate (like the elastic collision model), and prefers a residual connection when they are inaccurate (like the parabola equation).

**Failure Case** In differentiable NAS, the training algorithm aims to optimize the over-parameterized network with all the edges and operations. Therefore, it is necessary to prune the redundant edges and operations after training. We adopt the pruning mechanism in (Liu et al., 2018), where each node has to retain two incoming edges. For extreme cases where single-stream architectures are optimal, PhysicsNAS may generate sub-optimal architectures due to this arrangement. As shown in Figure 7, we make a toy comparison between two lightweight architectures on the collision task with the friction coefficient range $[0.15, 0.25]$ and 32 training samples. It is notable that by simply adding an additional stream to the input $x$, the new searchable architecture deteriorates the result. Introducing an adaptive edge selection mechanism might be a meaningful future work. This limitation could also be overcome by resorting to other NAS frameworks, such as those based on reinforcement learning (Pham et al., 2018).

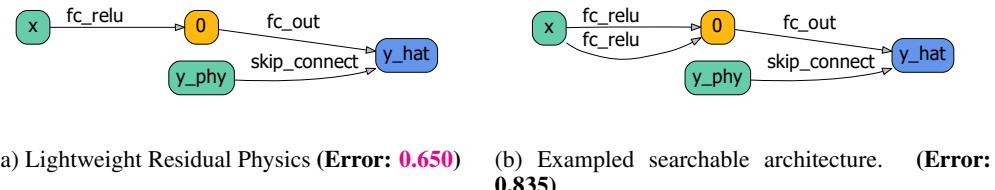

(a) Lightweight Residual Physics **(Error: 0.650)**     (b) Exampled searchable architecture. **(Error: 0.835)**

Figure 7: **Failure case.** In rare situations, a single-stream network could be preferred. PhysicsNAS is unable to converge to single-stream architectures due to the edge selection mechanism.

## 5 CONCLUSION

In conclusion, our experiments show that PhysicsNAS can handle a wider range of input physical models and data, as compared to existing PBL methods. This is only a first attempt at increasing the diversity of PBL through architecture search. Ultimately, our hope is to apply PhysicsNAS to problems as diverse as microscopy(Barbastathis et al., 2019), computer vision (Velten et al., 2012), sensor fusion (Eitel et al., 2015; Xu et al., 2018) and astrophysics (Bouman et al., 2016; Akiyama et al., 2019), where it is important to handle variations in model mismatch and dataset quality across these problem domains.

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

## A    GENERALIZABILITY FROM PHYSICAL PRIOR

We present a VISUAL PHYSICS task to further illustrate why physical prior is significant to data-driven approaches in a qualitative manner. The network tries to estimate the future trajectory of a paper ball being tossed from the initial three frames. We deploy similar physical prior as shown in Equation 3. This simple physical prior fails to generate reliable estimation due to the physical disparity from real world, such as the existence of air resistance and the deformation of the paper during traveling. From pure data-driven approach, we train a ResNet18 (He et al., 2016) with 32 tossing samples. This network still could not generate reliable estimation due to the limited number of training samples. Most of the predicted trajectories are not physically plausible, since the ball should not change its position suddenly in Frame 4. For the PBL method, we design a physics-informed ResNet18 with physical estimation as an additional input (Physical Fusion) and a skip connection between the physical estimation and the output (Residual Physics). The PBL network is also trained with 32 samples. With the assistance of physical prior, the trajectory could be estimated accurately. Some typical predictions of the above three methods are illustrated in Figure 8.

## B    COMPLETE LIST OF SEARCHED ARCHITECTURES

Searched architectures of PhysicsNAS in Table 1 and Table 2 are displayed in Figure 9 and Figure 10.

## C    EFFECTIVENESS OF EDGE WEIGHTS

We evaluate the effectiveness of edge weights, and exhibit the advantage of PhysicsNAS as compared to the original NAS framework. We conduct comparison between two different NAS policies: 1) the NAS framework with physical inputs and operations, yet without edge weights; 2) the proposed PhysicsNAS with both edge weights and physical modules. The initialized architectures

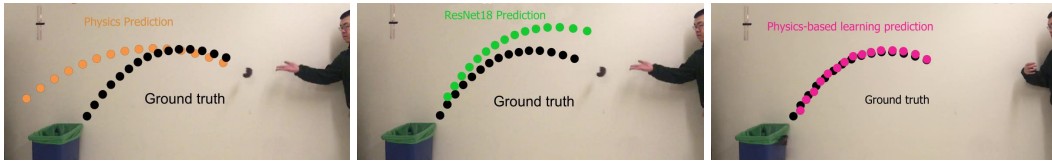

Figure 8: **Even from limited training samples, PBL has strong generalizability as compared with conventional data-driven approaches.** As exhibited in the left, estimation from physics can not match the ground-truth trajectory due to the physical disparity. As a data hungry method, deep neural networks also fail to generate physically plausible estimation with limited training samples (the middle). As depicted in the right of the figure, PBL method solves the dilemma by utilizing knowledge from both data and physical prior. The predicted trajectory (marked as green dots) from PBL highly overlaps with the ground truth in the testing stage.

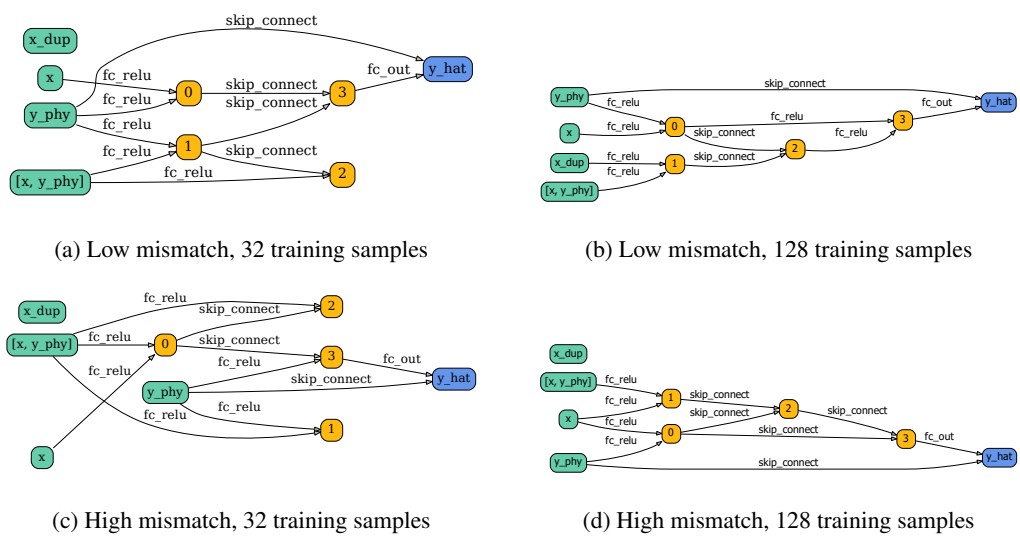

(a) Low mismatch, 32 training samples

(b) Low mismatch, 128 training samples

(c) High mismatch, 32 training samples

(d) High mismatch, 128 training samples

Figure 9: **Searched architectures of tossing task under diversified physical conditions.**

before neural architecture searching are illustrated in Figure 11, and the corresponding searched architectures are illustrate in Figure 12. We can observe that if the architectures are shallow in the initialization stage, there is a high probability that they remain shallow after searching. This phenomenon may come from the nature of differentiable NAS training process: once an operation is selected in a training update, it obtains privilege over other operations due to the backpropagation process, which in turn increases its weight and probability of being selected in the following training updates. Consequently, if the edges are selected merely based on the operation weights, NAS frameworks without edge weights will lose their potential to generate deep architectures due to the preference caused by difference in numbers of operations inside edges. PhysicsNAS addresses this issue by introducing additional edge weights, and the initial preference caused by unbalanced amount of operations within edges is thus alleviated. Comparison in Figure 11 and Figure 12 is conducted on the tossing task as detailed in Section 3.1. The range of random accelerations is $[-1m/s^2, 1m/s^2]$ and the damping factor is set to be 0.2. There are 128 training samples used for architecture searching and retraining.

## D EFFECTIVENESS OF TASK-SPECIFIC/PHYSICS-BASED ADAPTATIONS

We conduct an ablation study to demonstrate that the task-specific adaptations (i.e., adding physical inputs and operations into search space) in PhysicsNAS improve the predicted result. To this end, we conduct a comparison between: 1) the NAS framework with edge weights only; 2) the proposed PhysicsNAS with both edge weights and physics-based adaptations. The comparison is conducted

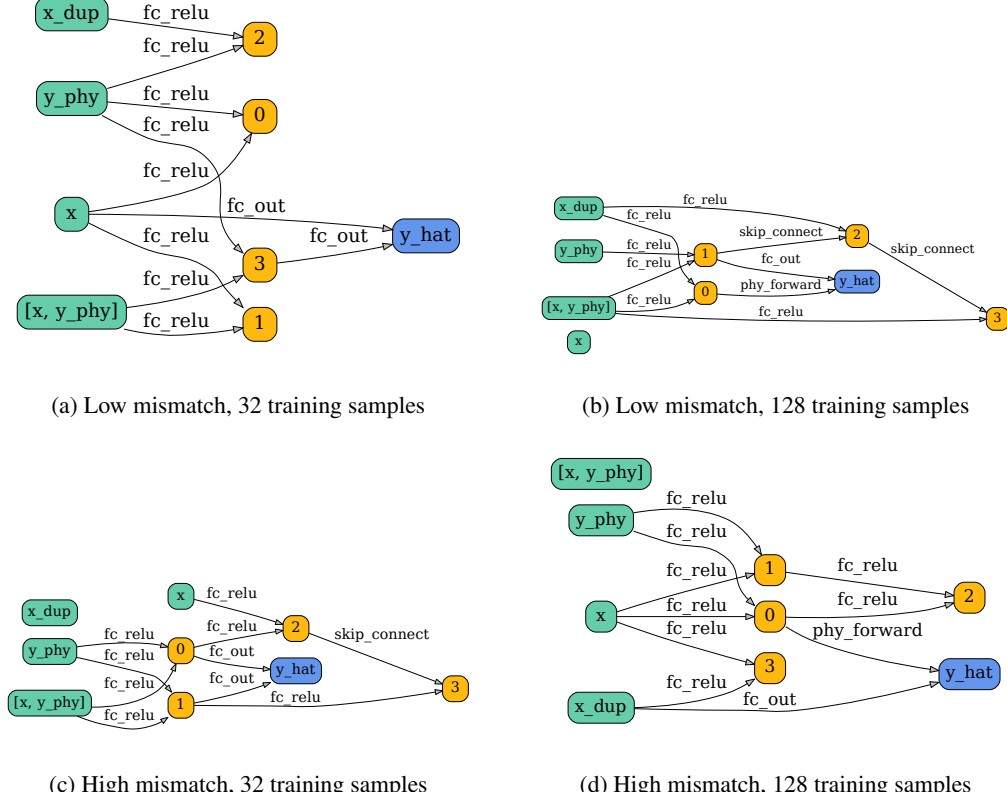

(a) Low mismatch, 32 training samples

(b) Low mismatch, 128 training samples

(c) High mismatch, 32 training samples

(d) High mismatch, 128 training samples

Figure 10: **Searched architectures of collision task under diversified physical conditions.**

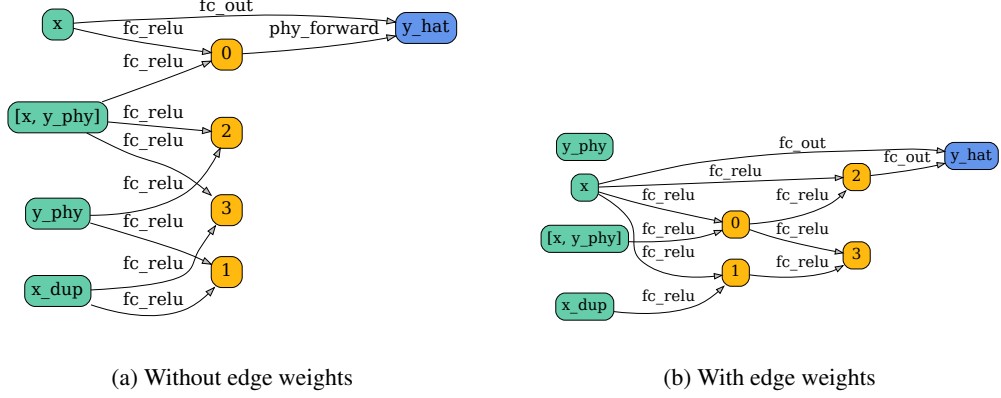

(a) Without edge weights

(b) With edge weights

Figure 11: **Initialized architectures of different NAS policies.** The initialized architecture of NAS without edge weights is generally shallow, since the number of the candidate operations between the input nodes and the hidden nodes is less than the number of candidate operations between hidden nodes themselves.

using the tossing task. As can be seen in Fig. 13, the introduction of physics-based adaptations enables PhysicsNAS to achieve significantly lower prediction error as compared to the NAS framework without task-specific adaptations.

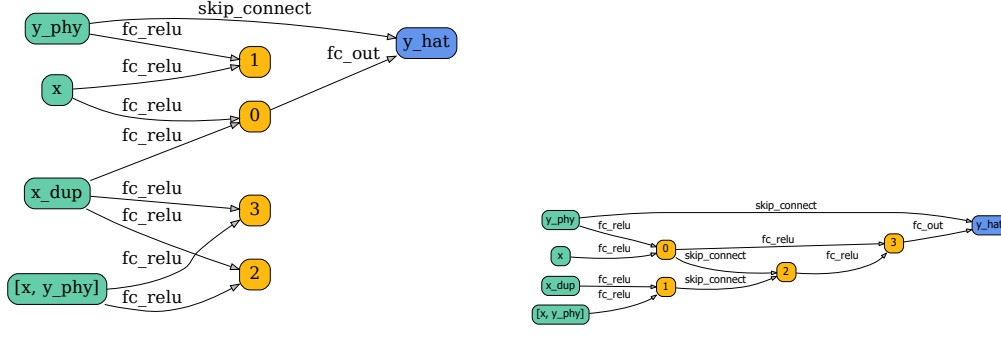

(a) Without edge weights **(Error: 0.205)**    (b) With edge weights **(Error: 0.097)**

Figure 12: **Searched architectures using edge weights alleviates model collapse.** The searched architecture without edge weights remains shallow due to the initial preference caused by unbalanced amount of operations, and its corresponding testing error is inferior to PhysicsNAS. The proposed PhysicsNAS has the potential to generate both deep and shallow architectures, while NAS without edge weights would generally converge to shallow architectures. The searched architecture of PhysicsNAS is thus superior to the one without edge weights in terms of the testing error.

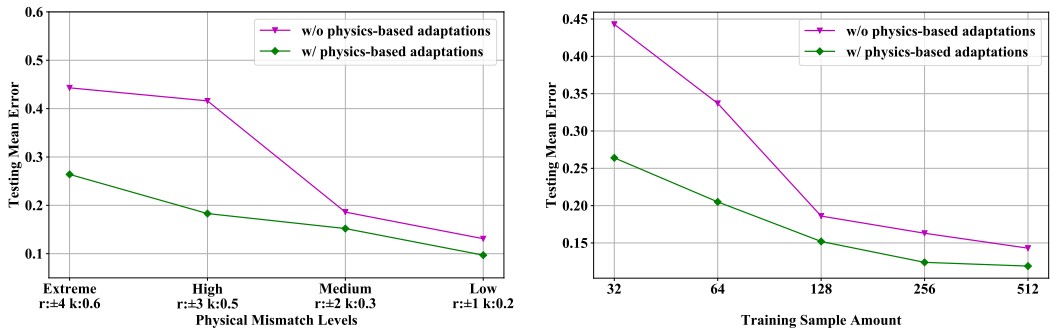

Figure 13: **PhysicsNAS has lower errors compared with NAS framework without physics-based adaptations over a range of quality conditions in physics and data.**

# E    DATA EFFICIENCY OF PHYSICSNAS

We show that the introduction of physical prior knowledge reduces the data amount for learning/approximating an exact physical model by comparing PhysicsNAS with naive MLPs along the data dimension. This comparison is conducted using the aforementioned tossing task in the high mismatch level scenario. As shown in Figure 14, PhysicsNAS can achieve significant, high prediction precision with limited data. As for this model mismatch type and level, PhysicsNAS can decrease the demand for training data by approximately 75%. This demonstrates the capability of PhysicsNAS on fewer-shot learning, where training data are burdensome to acquire due to extreme environments.

**Performance gain decreases as data amount increases.** As also shown in Figure 14, the performance gap between PhysicsNAS and the naive MLP continuously decreases as the amount of training samples increases. When the size of the training data is increased to 1024 the performance of the two models is nearly identical, with the MLP method slightly outperforming PhysicsNAS. This indicates that performance gain for applying PhysicsNAS is more evident when the amount of high-fidelity training data is limited.

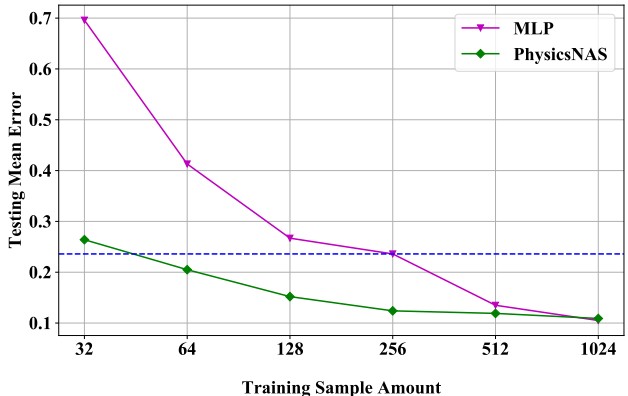

Figure 14: **Data Efficiency of PhysicsNAS.** With a rough physical prior, the demand of high-fidelity training samples can be greatly reduced. PhysicsNAS only requires less than 64 training samples to reach the testing performance of a Naive MLP with 256 training samples. It is reasonable to deploy PhysicsNAS to alleviate the burden of data acquisition when there are limited training samples.

## F ARCHITECTURE CHANGES AT DIFFERENT EPOCHS

We show the usage of physical operations along with the number of training epochs in this section. In Figure 15, the experiment is conducted in the tossing task with low mismatch level and 128 training samples, while Figure 16 shows the architecture changes in the collision task with high mismatch level and 128 training samples.

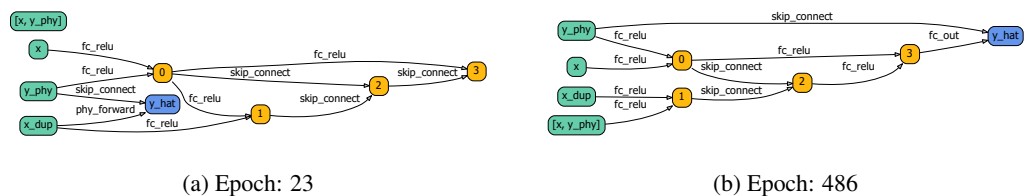

(a) Epoch: 23            (b) Epoch: 486

Figure 15: **Changes of searched topology in tossing task.** Inaccurate physical operation is used at early epochs, however, it is discarded as the training proceeds.

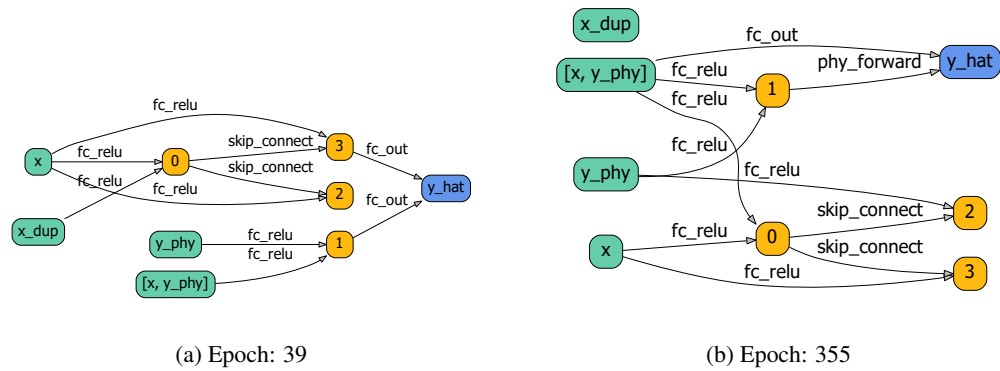

(a) Epoch: 39            (b) Epoch: 355

Figure 16: **Changes of searched topology in collision task.** The physical operation is not adopted initially, yet the searched topology eventually adopts the physical operation.

# G    UNCERTAINTY OF DIFFERENTIABLE ARCHITECTURE SEARCH

DARTS (Liu et al., 2018) framework tries to find an optimal architecture by updating the architecture parameters and network weights simultaneously. Therefore, the inconsistency between the architecture optimizer and the network weight optimizer may impede this joint optimization process. The performance of searched results relies on both the hyperparameters and the initialization, and improper training process may lead to model collapse as addressed in (Liang et al., 2019). PhysicsNAS is built based on the DARTS framework, and the model collapse issue also exists. Please refer to Figure 17 for some collapsed models of PhysicsNAS in the two demonstrative tasks. In our paper, we avoid the model collapse by tuning the learning rates of architecture and network optimizers, so that architecture parameters and network parameters can converge synchronously.

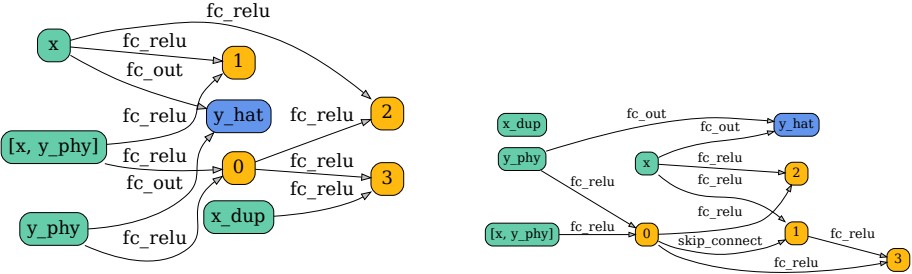

(a) Model collapse in the tossing task.          (b) Model collapse in the collision task.

Figure 17: **Model collapse of PhysicsNAS due to the uncertainty of training.** The performance of PhysicsNAS depends on the differentiable searching process. Model collapse may appear if hyperparameters are not selected carefully.

