# OpenReview forum: "Blending Diverse Physical Priors with Neural Networks"
_ICLR.cc/2020/Conference — Reject_

### Official Review · AnonReviewer3 · 2019-10-21
**Official Blind Review #3**

**Rating:** 6

**Review:**

=== Overall comments ===
This paper proposes to generalize approaches to physics-based learning (PBL) by performing network architecture search (NAS) over elements from PBL models found in the literature. This entails including physical inputs to the network and the incorporation of new operations to the NAS. I think the idea has merit and rather like it. However, there are several aspects of the work that could be improved. The technical novelty is small, as the extension of the  existing NAS models to handle physical inputs and a few new operators is relatively straightforward. The experiments, while well designed, only explore uninteresting toy problems. While I appreciate the necessity to explore the methods performance in a more controlled setting, a more impactful testbed would be more convincing. Another drawback of the evaluation is the lack of a proper statistical analysis of the results, given the small data and model sizes.


=== Relevance & Prior Work ===
+ The related work gives a good summary and categorization of prior work in physics-based learning
+ The problem (physics-based learning) is interesting and relevant to the community


=== Novelty & Approach===
+ application of NAS to physics based learning
+ incorporation of physics solutions as inputs into differentiable NAS
+ creation of physics-informed operation sets to merge physical models into network
- technical steps to merge NAS and PBL are relatively straightforward


=== Evaluation ===
Two representative physical simulations were chosen for evaluation, where elements of the physics model are intentionally omitted,  1) estimating trajectory of a ball in presence of wind and air resistance, and 2) a collision speed simulation where two objects collide, where sliding friction is not accounted for in the physics model.

The baselines consist of: a 3-layer MLP (data-driven), a 3-layer MLP with Physical Regularization, a 3-layer MLP with residual connection to the physics prediction, an MLP with two input branches, on for the data and one for the physics predictions (Physical Fusion), and the Embedded Physics model which estimates parameters for the physics modelu using a 3-layer MLP.

PhysicsNAS can combine elements of the baseline models, but the total number of nodes is limited to 5.

+ Experiments testing the dependence of the model on the numbers of samples and the strength of the physical inconsistencies were conducted. In both cases, PhysicsNAS outperformed the best specialized physics models.

- The chosen testbed tasks are toy problems. While these types of experiments are necessary to understand the performance of the model, it would have been interesting to see PhysicsNAS applied to a more impactful task

- Given the size of the networks and the training data, there is no reason why a more sophisticated statistical analysis of the results wasn’t performed (confidence intervals, t-test, p-value). Similarly, a more complete set of experiments with more sample amounts could be provided with little effort.


=== Clarity & Other Comments ===
- “precious nodes” -> previous nodes


**Experience Assessment:**

I do not know much about this area.

**Review Assessment: Checking Correctness Of Derivations And Theory:**

I assessed the sensibility of the derivations and theory.

**Review Assessment: Checking Correctness Of Experiments:**

I assessed the sensibility of the experiments.

**Review Assessment: Thoroughness In Paper Reading:**

I read the paper at least twice and used my best judgement in assessing the paper.

---

> ### Author Response · Authors · 2019-11-15
> **Response to Reviewer3**
>
> Thank you for your detailed review.
>
> Q1: Technical steps to merge NAS and PBL are relatively straightforward
> A1: This point is detailed in the general reply to all reviewers, where we describe in greater detail the non-obvious adaptations we have made to merge NAS with PBL. In the revision, we have experimentally shown in Figure 13 that these physics-specific adaptations boost the result.
>
> Q2: More “impactful” physics tasks instead of projectile motion and collisions
> A2: We specifically chose widely known physics tasks like projectile motion and collisions. This allows readers to access our paper and improve the algorithms, without the barrier to entry of knowing specialized physics. Although widely known, these tasks can also be made very challenging as we show in Fig. 4 by adding model mismatch. In crux, a kinematic equation with an unknown model mismatch can be a harder physics problem than a differential equation that perfectly describes the system.
>
> Q3: Experiments with more sample amounts.
> A3: We have added experiments with more training sample amount (1024). The new results are in the revised version of Fig.14. We find that when the amount of training data surpasses a certain point (like 512, 1024 in tossing task), the performance of PhysicsNAS is no longer better than the MLP method. This indicates PhysicsNAS is more preferred in fewer-shot learning, where training data are burdensome to acquire due to extreme environments.
>
> Q4: Statistical analysis.
> A4: We have conducted statistical analysis as a reference.
> The results are shown below. Regarding the P-value, PhysicsNAS performs the best.
>
>
>               |Naive Network |Physical Fusion | Embedded Physics |  Residual Physics | PhysicsNAS |
>
> P-value|      0.7637         |     0.9721          |        0.8793              |        0.9688           |    *0.9744       |
>
> * denotes the best.

---

> > ### Comment · AnonReviewer3 · 2019-11-15
> > **Response**
> >
> > Dear Authors,
> >
> > Thanks for your reply - I have a few comments:
> >
> > 1) Thanks for your clarification on the technical steps to adapt NAS for PBL, I think this is helpful
> > 2) As I mentioned in my review, I agree that these toy problems are helpful and necessary to include. My point is that it would strengthen the paper to have a problem with practical interest as well (but it's not absolutely necessary)
> > 3) It's good that you included these experiments. I agree that although the MLP is competitive with large numbers of samples, there are interesting cases where few samples are available
> > 4) Can you clarify some details - e.g. what is the data you compared and what statistical test you used?

---

> > > ### Author Response · Authors · 2019-11-15
> > > **Re: Response**
> > >
> > > Thank you for your comments.
> > >
> > > Comment 2:
> > > A2)  Thank you for your understanding. We are interested in more practical tasks such as physics-based imaging problems. We will extend PhysicsNAS to these tasks in future work.
> > >
> > > Comment 4:
> > > A4) The evaluation is conducted on tossing task with 32 samples under low physical mismatch levels.
> > > We derive t-test and P-value by conducting comparisons between network predicted trajectories and ground truth trajectories.  Furthermore, the values in the table are absolute averages of t-test/p-value results of all trajectory points.
> > > We use the function ‘ttest_ind’ in ‘scipy’ to do the evaluation.
> > >
> > > Below is an updated version of the table with t-test result inside.
> > >
> > > |Naive Network |Physical Fusion | Embedded Physics |  Residual Physics | PhysicsNAS |
> > > T-test	|      0.3077     	|    0.0350          |      0.1541              |         0.0391           |    *0.0321       |
> > >
> > > P-value|      0.7637         |     0.9721          |        0.8793              |        0.9688           |    *0.9744       |
> > >
> > > * denotes the best.

---

### Official Review · AnonReviewer2 · 2019-10-25
**Official Blind Review #2**

**Rating:** 3

**Review:**

The paper proposes PhysicsNAS, which proposes a method to automatically design architectures that
incorporate domain expertise from phyiscs-based models while also accounting for potential
mismatch between the model and real world due to unaccountable factors. While existing work seem
to incorporate such information via one of 4 standard ways (given on page 2), the proposed work
attempts to meld them so as to find the optimal combination for the problem at hand and the data
available.

While I don't see anything fundamentally wrong with the paper, I do not feel that the technical
contributions are substantial enough to warrant acceptance at ICLR.
More specifically, the methodological novelty is limited and the experimental evaluation only
evaluates the method on two fairly simple problems.

On a positive note, the authors have done a good job of illustrating the idea and have compared it
to most natural baselines. I also thought that the illustrations of the architectures found
for different sample sizes (in the Appendix) quite insightful.

I encourage the authors to pursue this line of work, but test this on more complex prediction tasks
where entirely model-based approaches are unreliable, and entirely black-box estimators are sample
inefficient. It also seems that the approach need not be confined to physics per se - in many
problems in chemistry, materials science etc. scientists are looking for ways to incorporate domain
expertise while accounting for model-mismatch.


**Experience Assessment:**

I do not know much about this area.

**Review Assessment: Checking Correctness Of Derivations And Theory:**

I assessed the sensibility of the derivations and theory.

**Review Assessment: Checking Correctness Of Experiments:**

I assessed the sensibility of the experiments.

**Review Assessment: Thoroughness In Paper Reading:**

I made a quick assessment of this paper.

---

> ### Author Response · Authors · 2019-11-15
> **Response to Reviewer2**
>
> Thank you for your detailed review. Regarding technical novelty in merging NAS with PBL. Incorporating physical priors required (non-obvious) modifications to differentiable NAS. We add Fig. 13 and Appendix Section D to show that these task-specific adaptations significantly boost the result.
>
> Regarding the complex tasks, we specifically chose widely known physics tasks like projectile motion and collisions. This allows readers to access our paper and improve the algorithms, without the barrier to entry of knowing specialized physics. Although widely known, these tasks can also be made very challenging as we show in Fig. 4 by adding model mismatch. In crux, a kinematic equation with an unknown model mismatch can be a harder physics problem than a differential equation that perfectly describes the system.

---

### Official Review · AnonReviewer1 · 2019-10-29
**Official Blind Review #1**

**Rating:** 6

**Review:**

The authors apply neural architecture search techniques to the problem of physics based learning. It is interesting because it cleverly tackles the challenge of manually designing priors and network architectures. The results are also impressive as the proposed method surpasses all the considered baselines. Despite of the above upsides, I have the following questions/concerns.
1. There is limited technical novelty as the entire method is mainly based on previous work on neural architecture search. Nevertheless, it might be helpful to have some ablation study to show the improvement of the task-specific adaptations presented in the paper, with which I believe this could be a good paper on the application side.
2. I'm curious about the performance of the baseline methods given the same amount of computation. For example, is it possible to perform intensive hyperparameter tuning for the baselines to also obtain improvement. It seems that the authors did not discuss the computational costs and whether different methods are compared given the same cost.

**Experience Assessment:**

I do not know much about this area.

**Review Assessment: Checking Correctness Of Derivations And Theory:**

N/A

**Review Assessment: Checking Correctness Of Experiments:**

I assessed the sensibility of the experiments.

**Review Assessment: Thoroughness In Paper Reading:**

I read the paper at least twice and used my best judgement in assessing the paper.

---

> ### Author Response · Authors · 2019-11-15
> **Response to Reviewer1**
>
> Thank you for your detailed review.
>
> Q1: Ablation Study (Comparison with NAS without Task-specific Adaptations)
> A1: We have added such a comparison. The added experiment results in Appendix Section D and Fig.13  justify that our task-specific adaptations significantly boost the performance. The details about task-specific adaptations and our novelty in merging NAS with PBL are in reply ‘General reply about Novelty in merging NAS with PBL’.
>
> Q2: Intensive hyperparameter tuning for the baselines.
> A1: In our previous submission, we have tuned the hyperparameters for perfecting the baseline performance. We now make it clearer by adding illustrations in ‘Training details’, Section 4.3. The illustrations are ‘Moreover, for all baseline approaches we compare in this paper, we fine-tune their hyperparameters in order to make fair comparisons. We choose three hyperparameter sets for each scenario and run five times for each method. We finally pick out the best result for each method.’

---

### Author Response · Authors · 2019-11-15
**General reply regarding Novelty in merging NAS with PBL**

We thank the reviewers for their comments.

Although our primary contributions are to PBL, it is worth noting that conventional differentiable NAS approaches (Liu et al 2018darts) are not designed to incorporate physical priors. Since all 3 reviewers have mentioned our novelty in the merging of NAS with PBL, we summarize our modifications in addition to the conventional NAS below.

*****************************************
Novelty in merging NAS with PBL
*****************************************
We are indeed inspired by existing NAS methods, but incorporating physical priors required (non-obvious) modifications to differentiable NAS [Liu2018, Cai2018]. We’ve added a new result figure to the paper showing that---without our tweaks---naive NAS is outperformed by PhysicsNAS at every testing configuration (cf. Figure 13).

i. Modifications in search space:
We considered using only operation weights as in Darts [Liu2018] and Proxyless [Cai2018], but finally had to add edge weights because we found that when introducing physical operations, the resulting imbalance of operation numbers between edges made the previous NAS methods more prone to search out a collapsed model.  We also considered utilizing unrolled optimization to construct the search space and adding physical operation in the middle layers of the search space but we had to rule them out because we found the physical tasks we implement here all have closed-form solutions and are more suitable for optimization without iterative solvers. Eventually, we converged on three physics-motivated modifications, described in ===Section 3.3=== of the original submission.

In the revision, we’d like to highlight the experimental value of these three modifications.  ===Figure 13=== of the revised submission demonstrates that PhysicsNAS outperforms NAS over all tested model mismatch configurations and training sample sizes. This experiment helps justify the value of our three major adaptations.

ii. Modifications in optimization strategy:
The optimization strategy in PhysicsNAS is essentially modified to be a mixture of Darts and Proxyless as we adopt their advantages respectively. We found the real-valued path-weights utilized in Darts made PhysicsNAS tend to abandon the physical operations that naturally prefer small output values. Thus we adopted the probabilistic path weights utilized in Proxyless. As well, we found that the selection of only one edge for every node in Proxyless was inadequate for aggregating physical information and adopted the two-edge selection scheme utilized in Darts.

---

### Decision · Program_Chairs · 2019-12-19

**Decision:**

Reject

**Comment:**

This paper constitutes interesting progress on an important topic; the reviewers identify certain improvements and directions for future work, and I urge the authors to continue to develop refinements and extensions.